# The Effect of Vitamin D Supplementation on Skeletal Muscle in the *mdx* Mouse Model of Duchenne Muscular Dystrophy

**DOI:** 10.3390/sports7050096

**Published:** 2019-04-26

**Authors:** Danielle A. Debruin, Nicola Andreacchio, Erik D. Hanson, Cara A. Timpani, Emma Rybalka, Alan Hayes

**Affiliations:** 1Institute of Sport and Health, Victoria University, Melbourne 3011, Australia; Danielle.debruin@live.vu.edu.au (D.A.D.); nicola.andreacchio@live.vu.edu.au (N.A.); edhanson@email.unc.edu (E.D.H.); cara.timpani@vu.edu.au (C.A.T.); emma.rybalka@vu.edu.au (E.R.); 2Australian Institute for Musculoskeletal Sciences (AIMSS), Melbourne 3021, Australia; 3Department of Exercise and Sport Science, University of North Carolina at Chapel Hill, Chapel Hill, NC 27599, USA; 4Melbourne Medical School, The University of Melbourne, Melbourne 3010, Australia

**Keywords:** 1,25(OH)2D: 1,25-dihydroxyvitamin D, 25(OH)D: 25-hydroxyvitamin D3, DMD: Duchenne Muscular Dystrophy, *mdx*: Duchenne Muscular Dystrophy Mouse Model, Skeletal muscle

## Abstract

Vitamin D (VitD) has shown to be beneficial in reversing muscle weakness and atrophy associated with VitD deficiency. Duchenne muscular dystrophy is characterized by worsening muscle weakness and muscle atrophy, with VitD deficiency commonly observed. This study aimed to investigate the effect of VitD supplementation on dystrophic skeletal muscle. Eight-week old female control (C57BL/10; n = 29) and dystrophic (C57BL/*mdx*; n = 23) mice were randomly supplemented with one of three VitD enriched diets (1000, 8000 & 20,000 IU/kg chow). Following a four-week feeding period, the extensor digitorum longus (EDL) and soleus muscles contractile and fatigue properties were tested *ex vivo*, followed by histological analysis. As expected, *mdx* muscles displayed higher mass yet lower specific forces and a rightward shift in their force frequency relationship consistent with dystrophic pathology. There was a trend for *mdx* muscle mass to be larger following the 20,000 IU/kg diet, but this did not result in improved force production. Fiber area in the EDL was larger in *mdx* compared to controls, and there were higher amounts of damage in both muscles, with VitD supplementation having no effect. Four weeks of VitD supplementation did not appear to have any impact upon dystrophic skeletal muscle pathology at this age.

## 1. Introduction

Vitamin D (VitD) is a secosteroid that is essential for the proper functioning of the human body [1,2,3]. Most well-known for its role in regulating bone mineralization [4], VitD has many regulatory actions. In recent years, a link has been established between VitD and skeletal muscle suggesting that VitD may have beneficial effects on muscle strength, reducing muscle fibre atrophy, aiding in muscle regeneration following damage [5,6] and improving calcium handling [7,8,9]. These recent discoveries open up the possibility for therapeutic use of VitD in degenerative muscle disorders.

One of the most notable degenerative muscular disorders is Duchenne Muscular Dystrophy (DMD); an X-linked recessive disorder that affects 1 in 3500 live male births [10]. Characterized by progressive muscle weakness and gait disturbances [11], DMD is caused by the absence of the integral cytoskeleton protein dystrophin [12,13]. Diagnosed in early childhood, the disease progresses over time and leads to severe degeneration and wastage of skeletal muscle; typically resulting in death by the third decade of life [10]. Despite advances in the management of the disease and increased life expectancy, the irreplaceable loss of functional tissue and replacement with fat and connective tissue still leads to the dependence on a wheelchair for movement within a decade of being diagnosed [10,11,14]. The overall low quality of life in these patients highlights the need for new therapies that target muscle strength loss. The cycles of degeneration and incomplete regeneration, which are hallmark features of DMD, include some characteristic features of aging muscle, most specifically relating to loss in muscle strength and atrophy of type II muscle fibers [15]. VitD supplementation proves to be beneficial to muscle function in the elderly who are VitD deficient [16,17], as well as muscle atrophy conditions such as post-menopausal associated sarcopenia and cachexia in prostate cancer patients [18,19]. It is possible that VitD supplementation may have the same beneficial effects on dystrophic muscle as has already been shown in aging and VitD deficiency.

There are many preclinical animal models of DMD, including the most commonly used *mdx* mouse [20,21]. The *mdx* mouse is genetically homologous to the human condition and as a result, many of the pathophysiological processes involved in the damage/regeneration cycle as seen in the human condition are present. Characteristic features of human DMD that are also present in the *mdx* mouse model include centralized nuclei (an indicator of muscle regeneration), fiber size variability, muscle fiber atrophy and fiber necrosis and fibrosis [22]. As seen in the human disease, the *mdx* mouse also demonstrates Ca^2+^ dysregulation and persistent increased intracellular Ca^2+^ levels [23,24,25,26]. In the *mdx* mouse, muscle degeneration is not evident until three weeks of age after which time striated muscle undergoes severe degeneration [22,27]. At five weeks, damaged muscle enters a period of rapid regeneration which continues to approximately eight weeks of age, after which the hind-limb muscles of the *mdx* mice cease moving through cyclic phases of muscle degeneration and regeneration [20,28]. From eight-weeks onwards, the hind limb muscles of the *mdx* mouse remain in a state of low-grade chronic inflammation, damage and regeneration, and necrosis that persists through to adulthood [29]. Thus, to avoid the peak damage/regeneration period, this study aimed to investigate any beneficial effects on skeletal muscle function in eight-week old *mdx* mice. It was hypothesised VitD supplementation would have several effects on *mdx* dystrophic muscle including increased absolute force production, decreased fatigue following repeated muscle stimulation, increased recovery capacity post-fatigue and enhanced muscle regeneration.

## 2. Materials and Methods

### 2.1. Animals and Ethics Approval

Ethics approval was granted for this study from the Victoria University Animal Experimentation Ethics Committee (AEETH 13/007) and all experiments performed in relation to this study abided by the Australian Code of Practice for the Care and Use of Animals for Scientific Purposes. A total of 52 eight-week old female control (C57BL/10; n = 29) and dystrophic (C57BL10/*mdx*; n = 23) mice were obtained from the Animal Resource Center (ARC, Western Australia). Animals were chosen at this age to avoid the peak damage/regeneration period that occurs from 3–6 weeks of age, thus aligning with a period of slower progression of the disease occurring during a period of animal growth and maturation. Mice were randomly provided one of three chow diets varying in their VitD (cholecalciferol) content (Specialty Feeds, WA, Australia); referred to as low, medium and high. For the purpose of this study, the low diet (containing 1000 IU/kg chow) served as the control diet, as this is the standard content of VitD found in rodent chow and provides the equivalent of 400–1000 IU VitD per day in humans, which are fairly typical daily supplementation doses. The medium diet contained 8000 IU VitD per kilogram of chow whilst the high diet contained 20,000 IU/kg. Other than the differing VitD levels, the three diets were identical. Chow was contained within a single hopper in each cage and was made available to mice ad libitum following a three-day acclimatization period. Following a four-week feeding period, animals were anesthetized with an intraperitoneal injection of pentobarbitone sodium (60 mg/kg). Muscle tissue was then extracted and tested for functional and histological analysis.

### 2.2. Contractility

The right hind-limb extensor digitorum longus (EDL) and soleus muscles were excised, tendon-to-tendon and tied to a sensitive force transducer in a custom—built organ bath (Zultek Engineering) filled with Krebs-Henseleit Ringer’s solution bubbled with carbogen (5% CO_2_ in O_2_; BOC gases, Melbourne, Australia) maintained at a pH of 7.4 at 30 °C. Supramaximal square wave pulses (0.2 msec duration) were used to stimulate all muscle fibers and the force generated measured. The length at which the muscle was able to produce its greatest force was established and maintained for the duration of the testing protocol. In order to determine maximum absolute force (Po), muscles were stimulated at an increasing range of frequencies (10, 30, 50, 80, 100, 120, 150 and 180 Hz) with a pulse train of 350 and 500 msec for the EDL and soleus, respectively, with three-minute recovery between each stimulus to prevent fatigue.

### 2.3. Fatigue and Recovery

Following the determination of absolute force (Po), muscles were fatigued through the delivery of repeated, intermittent stimuli for a total duration of three minutes. The EDL was stimulated at 100 Hz every four seconds, whilst the soleus was stimulated at 80 Hz every two seconds in order to achieve a comparable level of fatigue in each of the muscles. Muscles were followed during recovery for 60 min from fatigue, during which time they were stimulated tetanically at specific intervals (1, 2, 5, 10, 15, 20, 30, 45 and 60 min) and the forces generated were recorded. Following the conclusion of the protocol, muscles were removed from their bath and cut free from tendons before being blotted to remove excess fluid from the bath and weighed. The cross-sectional area (CSA) for each muscle was determined using both the optimal length and mass of each muscle, according to the equation by Brooks & Faulkner [30] with muscle density assumed to be 1.06 g/cm^3^ [31]. Specific force (sPo) of each muscle was calculated by Po/CSA.

### 2.4. Histological Analysis

Contralateral EDL and soleus muscles were snap frozen in isopentane cooled in liquid nitrogen, and later cut at 12 μm using a cryostat (Leica, CM1950) and maintained at a temperature of −15 °C before being mounted on glass slides. Samples were stained for haematoxylin and Eosin (H&E) and photographed using a monochrome camera (Meta Systems, Cool Cube 1) connected to a microscope (Zeiss Imager.Z2). Sections were photographed at ×20 magnification. For analytical purposes, a further ×20 zoom was applied to three random separate areas using the Meta Systems Client Viewer program and the total number of muscle fibers located within each frame was counted, their area measured and averaged. The number of centrally located nuclei (indicative of regeneration) was then counted and expressed as a percentage of total nuclei, while the percent of the section exhibiting muscle damage (infiltrate and phagoctiyc material) was also analysed. 

### 2.5. Statistical Analysis

The data presented is expressed as mean ± SEM. A two-way analysis of variation (ANOVA) was used to analyze data with diet being the within-group factor and strain the between-group factor. If differences and/or interactions were observed Tukey’s post hoc test was employed to identify the location of differences. A *p* value of 0.05 or less was deemed statistically significant.

## 3. Results

Body weight and muscle characteristics are shown in Table 1. *Mdx* mice were significantly heavier than their healthy counterparts. Similarly, EDL and soleus muscle mass was significantly higher in *mdx* mice. Interestingly, although there was no significant direct effect of VitD, there did appear to be a higher sensitivity to VitD in the dystrophic muscles. The level of significance increased with each dose of VitD, such that despite no difference in muscle mass to body mass ratio with the control diet, the relative EDL mass was higher in *mdx* mice with both the medium (*p* < 0.05) and high (*p* < 0.001) diets compared to the control strain (see Table 1). A similar effect was seen in the soleus with the high diet (*p* < 0.05).

Despite this apparent effect, there was no translation into improved function of dystrophic muscles. As expected, *mdx* EDL muscles had both lower absolute and specific forces than their control strain counterparts (see Figure 1), due to preferential loss of fast-twitch fibers in dystrophic pathology. While soleus (slow-twitch) muscles had similar absolute forces, their larger size meant that specific forces was also lower in *mdx* mice compared to controls (see Figure 1). As mentioned, VitD had minimal effect on force production in either muscle, with higher specific forces (*p* < 0.05) in the soleus muscle with the high diet compared to low, but only in the control strain. 

Force-frequency relationships demonstrated significant rightward shifts (*p* < 0.001) in the *mdx* mice in both the EDL and soleus muscles compared to the control strain (see Figure 2). VitD supplementation again had no significant effects, although there was a trend for higher forces at lower frequencies after the high diet in the EDL muscle.

Despite no effect of VitD on force development, it appeared that medium and high doses of VitD improved fatigue in the EDL of the healthy animals (*p* < 0.05, Figure 3A). Furthermore, there was a strong tendency (*p* < 0.1) in the healthy high dose EDL muscles to be higher than the healthy low VitD supplemented during recovery (see Figure 3). Conversely, no direct effect of VitD was found on fatigability or recovery in the *mdx* EDL muscles - only strain differences were observed during the later stages of recovery between the medium diet groups (*p* < 0.05, Figure 3B). Interestingly, no strain differences were observed in the soleus muscles during the fatigue and recovery protocols, with only the healthy medium diet animals displaying lower rates of recovery at 15 and 30 min when compared to the high group (*p* < 0.05, Figure 3C). There was also a strong tendency for the high dose non-dystrophic muscles to have a higher recovery capacity than the medium group at 20 and 45 min (*p* < 0.08 and *p* < 0.06, respectively). However, no such effects were observed in the soleus muscles of *mdx* mice.

Analysis of fiber size distribution in EDL muscles stained with H&E found that there was no significant effect of VitD in either the non-dystrophic or *mdx* mice (see Figure 4A). As expected, there was increased variation in frequency of fibre distributions in both EDL and soleus muscles of *mdx* mice (see Figure 4 and Figure 5), reflective of damage, regeneration and pseudohypertrophy found across all VitD diets (*p* < 0.0001; Figure 4D). Interestingly, CSA was unaffected in the slow-twitch soleus muscle with the strain and VitD having no impact. Analysis of control muscle sections found there were no central nuclei located in samples, thus regenerative capacity was analysed only in the *mdx* muscle samples. As expected, both EDL and soleus *mdx* muscles demonstrated high proportions of regenerating fibres (~30–40%), and this was unaffected by VitD (Figure 4E and Figure 5E). Phagocytic infiltration of muscle was evident in both EDL and soleus *mdx* muscle samples. These areas depict phagocytic cells converged at various sites of muscle damage and were analysed as a percentage of total area (Figure 4F and Figure 5F). Again, there were no differences with VitD supplementation.

## 4. Discussion

The major finding of the current study was that VitD supplementation did not appear to have a significant impact on either contractility or regeneration in *mdx* mouse muscles. Impaired muscle function in the *mdx* mouse was as expected, with deficits in force production and calcium handling, which was accompanied by instances of muscle fiber damage and regeneration. Indeed, force production of dystrophic muscle is increasingly affected as the number of degenerating muscle fibers and regenerating myotubes increases over time as result of the genetic-mutation that leads to the loss of the protein dystrophin [32,33,34,35], resulting in lower specific forces in dystrophic muscles. 

In the human form of the disease, boys with DMD lose the ability to ambulate during the teenage years as muscle damage and loss of functional tissue leads to an inability to generate appropriate amounts of force. In this present study, 12-week old EDL and soleus muscles from *mdx* animals had significantly lower specific forces than the controls regardless of diet (see Figure 1). Moreover, the clear impact of the dystrophic condition on type-II fibers are illustrated in the EDL of *mdx* mice, which produce significantly less absolute force than healthy controls despite an increase in the size of their muscles, something which is commonly seen in dystrophic fast-twitch muscles [36], resulting in lower specific force. Whilst muscle fibers are damaged and lost, it is believed that surrounding fibers compensate for this and hypertrophy [15] and is indicated by an increase in muscle mass size relative to non-dystrophic controls (see Table 1). However, while this maintains force in slow-twitch muscles such as the soleus, the amount of regenerated and non-contractile material also results in lower specific force. 

The presence of a rightward shift in the force-frequency relationship can indicate a shift to a faster phenotype, as higher frequencies of activation are required for summation. However, lower forces and preferential loss of fast-twitch fibers in dystrophic muscles [37,38,39] make that explanation unlikely. As such, lower forces displayed by the *mdx* mice suggest changes in Ca^2+^ sensitivity, as a higher frequency (to elicit sufficient calcium release) is required to produce maximal force. While altered calcium dynamics have been well reported in dystrophic muscle [40,41,42], there does not appear to be any effect of VitD on the force frequency relationship, although further studies with single muscle fibers are required to fully elucidate this. 

As well as no effect on the force frequency (and thus calcium handling), there was also no effect of VitD on muscle mass or force, although there was some suggestion of heightened sensitivity to VitD in the dystrophic muscles. This was despite previous research demonstrating that VitD may aid muscle recovery and improve muscle strength [5,6,7]. Further, there were also limited effects on fatigue and recovery with VitD supplementation. The lack of VitD effect in the dystrophic EDL muscles may be due to the gradual increase in the proportion of fatigue-resistant slow-twitch fibers [43,44]. It may be that effects of VitD are masked by the higher fatigue resistance offered by slow-twitch muscle fibres. Indeed, the *mdx* EDL did not fatigue to the same level as control EDL muscles after two minutes and then again at multiple points during recovery (see Figure 3). In support of this, beneficial effects of VitD were not observed in the non-dystrophic soleus muscles suggesting that effects are only provided to fast-twitch fibers. Since there is preferential loss of fast-twitch fibers in dystrophic EDL muscles, this may explain why effects are only seen in the non-dystrophic animals, and suggests that any VitD therapy would need to be started prior to significant fast-twitch fiber loss.

The data from this study does not support the use of VitD supplementation as it did not lead to increased muscle force output or return dystrophic EDL or soleus muscle forces to control levels. However, there were a number of limitations to the study. VitD may be most effective during periods of damage and repair [5]. The current study deliberately avoided the peak damage and repair period as this is well above what is observed in human dystrophy. As a result, the study was during a period of lower levels of muscle degeneration and regeneration. As such, there may have been insufficient stimulus for VitD to have any major effect, or insufficient fast-twitch fibres for any effect to be realised. Whilst muscle degeneration can be equally matched with regeneration for a sustained amount of time in the *mdx* mouse, degeneration slows considerably, and only low-grade chronic inflammation persists [29]. Thus, it is possible that in this study, the damage-regeneration phase from 8- to 12-weeks is not severe enough for VitD to take effect. Alternately, if VitD were decreasing the number of degenerating muscle fibers, whilst also increasing the rate of regeneration, as suggested by [5,45] then the overall outcome in centralised nuclei could be the same and may conceal any effects of VitD. It is reasonable to suggest that future studies should consider timing of supplementation and apply it during the peak damage period between 14–28 days old. This is even more pronounced when it is considered that VitD supplementation has been shown to be most effective in those deficient in VitD. While DMD sufferers are more likely to be VitD deficient, the current study investigated mice that would have had usual levels of VitD, as they were on the typical (1000 IU/kg chow) diet until the study began at 8 weeks of age. Thus, it is possible that the lack of effect in the dystrophic animals may be due to the animals beginning supplementation with normal levels of VitD. If the *mdx* mice were deficient prior to VitD feeding, the severity of the dystrophic phenotype would likely increase, thus enabling VitD to have a more significant effect. Indeed, studies have demonstrated that VitD deficient non-dystrophic mice are weaker than sufficient controls and is effectively reversed with VitD supplementation [7]. 

Despite no direct impact of VitD on muscle function in the *mdx* mouse, it appeared to have improved force production in the soleus and fatigue resistance in the EDL in normal muscles. The ergogenic potential of VitD to increase muscle performance has been discussed previously [46]. The studies outlined demonstrated a positive link between VitD status and various aspects of muscle function that athletes require for optimal performance. For example, it has been found in numerous studies that serum 25(OH)D levels are positively correlated with muscle force, power, velocity and jump height [47,48,49,50]. In addition to this, a study investigated whether serum VitD levels could influence the recovery rates of muscle strength after resistance training-induced muscle damage [51]. It was found that pre-exercise VitD serum levels predicted the level of recovery of muscle strength after an acute bout of resistance exercise. Specifically, the higher the level of circulating VitD in the blood [25(OH)D] prior to exercise, the lower the post-exercise muscle weakness. By improving the recovery time from intense exercise, this would potentially allow athletes to train more intensely and/or more frequently, which is likely to increase exercise performance over a long period of time.

Increased recovery rates would also benefit aerobic exercise performance, with a link between aerobic athletic performance and VitD status in humans made via correlative studies [52,53]. A positive correlation between VitD status and VO_2_ max (the gold standard indicator of aerobic cardiorespiratory fitness) [54] suggest that subjects with a high VitD status have increased aerobic capacity and, thus, that VitD supplementation may potentiate exercise abilities. However, most VitD supplementation studies have been conducted on VitD deficient or insufficient athletes, in which the greatest improvements in muscle strength, power and speed are seen compared to their control counterparts with normal endogenous VitD levels (>50 nmol/L) [48,49,50]. Furthermore, there is a lack of research as to whether VitD supplementation can improve exercise performance through decreasing limiting factors such as muscular fatigue and damage in athletes who have sufficient levels of VitD. Thus, future studies that combine high dose VitD supplementation and exercise in healthy models should be considered.

In the current study, it was observed that the control animals fed the high VitD diet elicited greater forces per CSA when compared to the low and medium diets (Figure 1D). Furthermore, supplementation with the medium and high VitD diets delayed fatigue during the first two minutes of fatigue in non-dystrophic EDL muscles. These findings suggest that increased VitD (from a non-depleted state) may aid in conserving force output in slow-type I muscle fibers and aids in improving fatigue and recovery in the highly fatigable type-II fibers. However, further work is required to investigate this.

## 5. Conclusions

In conclusion, VitD supplementation did not appear to have any beneficial effect on the contractile properties or regenerative capacity of dystrophic skeletal muscle in the *mdx* mouse. Following contractile testing, it was observed that an increase in VitD intake did not lead to increased force output in dystrophic compared to non-dystrophic animals. Further to this, fatigability of dystrophic muscles and their ability to recover from fatigue was also not affected by VitD supplementation. Beginning supplementation earlier, or from a VitD deficient state, may more definitively answer whether there really is no effect. Although differences were not observed as a result of VitD supplementation in the *mdx* animals, several differences were observed in the maintenance of force during fatigue and recovery in control animals. These findings could suggest that VitD supplementation may be beneficial from an athletic point of view to increase muscle performance when applied in a VitD replete state, thus providing a further avenue for future research.

## Figures and Tables

**Figure 1 sports-07-00096-f001:**
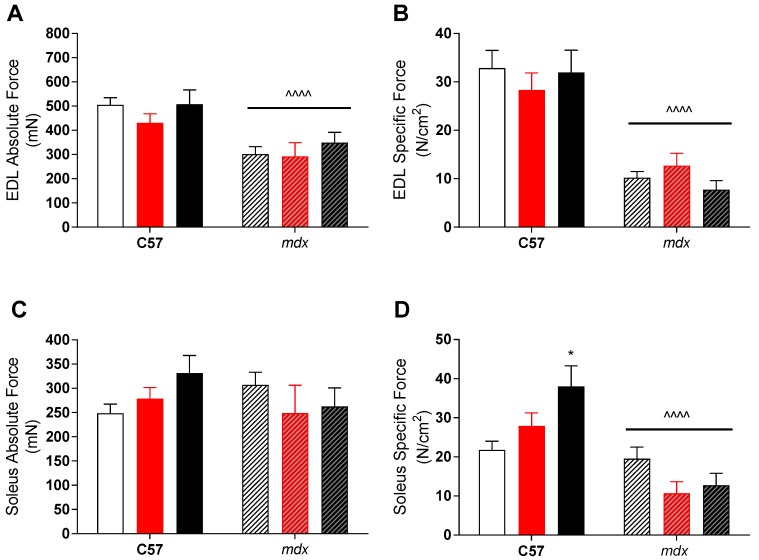
Extensor digitorum longus (EDL) and soleus (SOL) absolute and specific force production in non-dystrophic and *mdx* animals. Absolute tetanic force (**A**) and specific force (**B**) of the EDL muscle from C57 and *mdx* mice from low (open bar; n = 4–10), medium (red bar, n = 6–10) and high (black bar; n = 6–10) vitamin D enriched diets. SOL absolute (**C**) and specific force (**D**) from C57 and *mdx* mice from low (open bar; n = 6–10), medium (red bar, n = 6–10) and high (black bar; n = 6–10) vitamin D enriched diets. Symbols indicate: * *p* < 0.05, significantly different to low diet. ^^^^ *p* < 0.0001, significantly different to C57 strain.

**Figure 2 sports-07-00096-f002:**
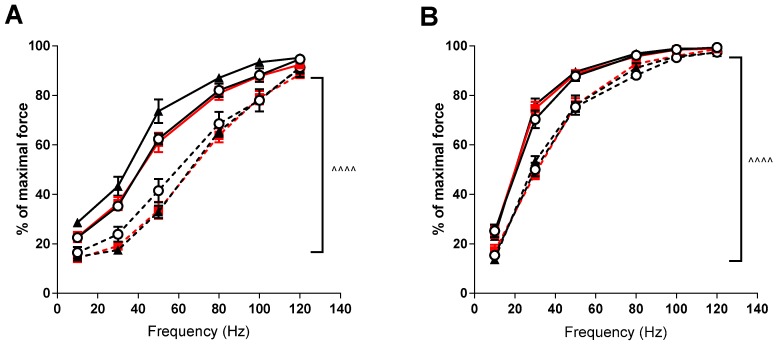
EDL and soleus (SOL) force frequency relationship in non-dystrophic and *mdx* animals. EDL muscles (**A**) from C57 mice fed low (**⭘**; n = 5), medium (■; n = 4) and high (⯅; n = 6) vitamin D enriched diets and *mdx* mice fed low (**⭘**, broken line; n = 6), medium (■, broken line; n = 6) and high (⯅, broken line; n = 5) vitamin D enriched diets. SOL muscles (**B**) from C57 mice fed low (**⭘**; n = 5), medium (■; n = 4) and high (⯅; n = 6) vitamin D enriched diets and *mdx* mice fed low (**⭘**, broken line; n = 6), medium (■, broken line; n = 6) and high (⯅, broken line; n = 5) vitamin D enriched diets. Symbols indicate: ^^^^ *p* < 0.0001 significant difference between C57 and *mdx* strains.

**Figure 3 sports-07-00096-f003:**
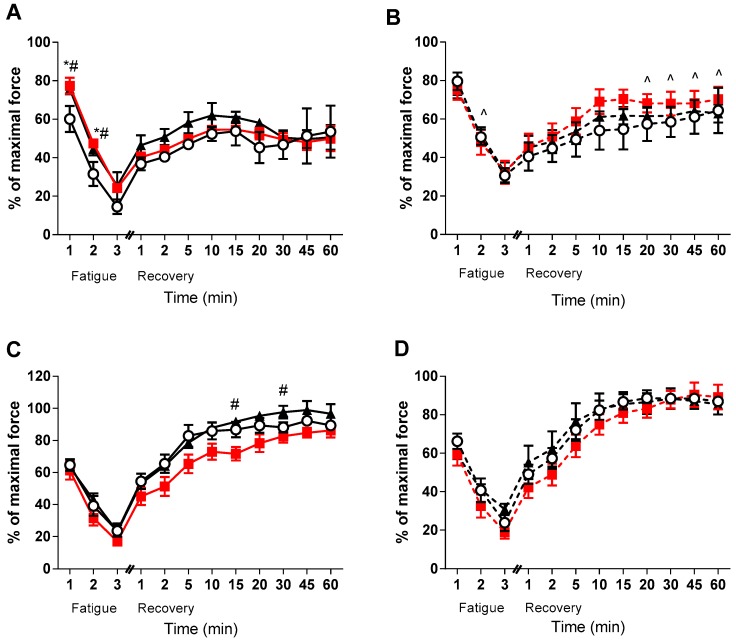
EDL and soleus (SOL) Fatigue and Recovery in non-dystrophic and *mdx* animals. Fatigue and Recovery of (**A**) EDL muscles from C57 mice from low (**⭘**; n = 5), medium (■; n = 4) and high (⯅; n = 6) vitamin D enriched diets. (**B**) EDL muscles from *mdx* mice from the low (**⭘**, broken line; n = 6), medium (■, broken line; n = 6) and high (⯅, broken line; n = 5) vitamin D enriched diets. (**C**) SOL muscles from C57 mice from low (**⭘**; n = 5), medium (■; n = 4) and high (⯅; n = 6) vitamin D enriched diets and (**D**) SOL muscles from *mdx* mice from the low (**⭘**, broken line; n = 6), medium (■, broken line; n = 6) and high (⯅, broken line; n = 5) vitamin D enriched diets. Symbols indicate: * *p* < 0.05, significantly different from low diet). # *p* < 0.05, significantly different from the medium diet. ^ *p* < 0.05, significantly different from the C57 strain.

**Figure 4 sports-07-00096-f004:**
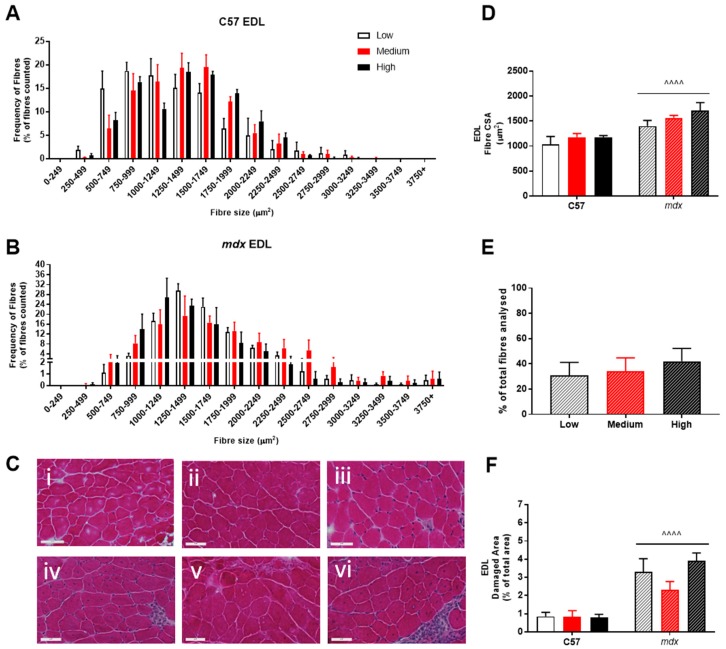
Histological analysis of the EDL in non-dystrophic and *mdx* animals. Fibre size distribution in non-dystrophic (**A**) and *mdx* (**B**) animals treated with low (white bar), medium (red bar) and high (black bar) vitamin D enriched diets. Representative images are depicted in panel (**C**); low, medium and high vitamin D diets (C57 i–iii and *mdx* iv–vi, respectively). Fibre cross-sectional area (**D**), percentage of regeneration (**E**); *mdx* only) and damage (**F**) was also analysed in non-dystrophic C57 and *mdx* mice fed low, medium and high amounts of vitamin D. Symbols indicate: ^^^^ *p* < 0.0001, significantly different to C57 strain. All groups n = 4. Scale bar = 50 μm.

**Figure 5 sports-07-00096-f005:**
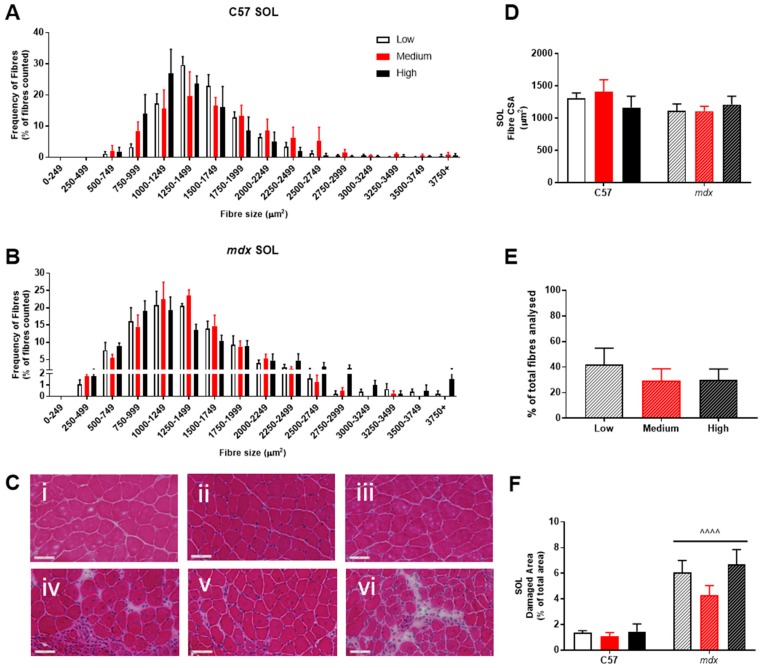
Histological analysis of the soleus (SOL) in non-dystrophic and *mdx* mice. Fiber size distribution in non-dystrophic (**A**) and mdx (**B**) animals treated with low (white bar), medium (orange bar) and high (black bar) vitamin D enriched diets. Representative images are depicted in panel (**C**); low, medium and high vitamin D diets (C57 i–iii and *mdx* iv–vi, respectively). Fibre cross-sectional area (CSA; (**D**)), percentage of regeneration (*mdx* only; (**E**)) and damage (**F**) was also analysed in non-dystrophic C57 and *mdx* mice fed low, medium and high amounts of vitamin D. Symbols indicate: ^^^^ *p* < 0.0001, significantly different to C57 strain. All groups n = 4. Scale bar = 50 μm.

**Table 1 sports-07-00096-t001:** Body mass and muscle morphology characteristics.

Measure		C57BL/10			*mdx*		
		1000 IU n = 10	8000 IU n = 10	20,000 IU n = 9	1000 IU n = 7	8000 IU n = 8	20,000 IU n = 8
Body Weight (g)		22.1 ± 1.3	22.2 ± 1.6	23.9 ± 0.97	24.6 ± 0.5 ^^^	25.2 ± 1.0 ^^^	25.5 ± 1.1 ^^^
EDL	Mass (mg)	9.5 ± 0.6	8.7 ± 0.4	8.8 ± 0.3	16.8 ± 2.2 ^^^	17.3 ± 3.4 ^^^^	23.1 ± 1.8 ^^^^^
	EDL:BM (mg/g)	0.046 ± 0.002	0.041 ± 0.002	0.041 ± 0.002	0.068 ± 0.009	0.074 ± 0.015 ^^^	0.090 ± 0.007 ^^^^^
	CSA (cm²)	0.017 ± 0.001	0.016 ± 0.001	0.016 ± 0.001	0.035 ± 0.006 ^^^	0.038 ± 0.008 ^^^^	0.047 ± 0.004 ^^^^^
SOLEUS	Mass (mg)	8.75 ± 0.47	7.42 ± 0.42	7.87 ± 0.43	14.9 ± 1.7 ^^^	14.8 ± 1.2 ^^^^	15.7 ± 2.4 ^^^^^
	Soleus:BM (mg/g)	0.041 ± 0.002	0.035 ± 0.002	0.038 ± 0.003	0.061 ± 0.007	0.059 ± 0.005	0.072 ± 0.018 ^^^
	CSA (cm²)	0.010 ± 0.001	0.010 ± 0.001	0.009 ± 0.001	0.017 ± 0.002 ^^^	0.021 ± 0.002 ^^^^^	0.018 ± 0.002 ^^^^

Abbreviatons: EDL = extensor digitorum longus; BM = body mass; CSA = cross sectional area. Symbols indicate: ^ *p* < 0.05, ^^ *p* < 0.01, ^^^ *p* < 0.001; Significantly different from C57 strain.

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
