# Peer review of "The Effect of Vitamin D Supplementation on Skeletal Muscle in the mdx Mouse Model of Duchenne Muscular Dystrophy"

_sports, 2019, doi:10.3390/sports7050096_

Reviewer 1 Report

This study sought to examine the effect of Vitamin D supplementation on a mouse model of Duchenne muscular dystrophy. Primary findings indicate that none of the three Vitamin D enriched diets (1000, 8000 or 20,000 IU/kg chow) have beneficial effects on skeletal muscle function of mdx mouse. Paper is well written, and data are presented in a clear and logical manner.

I have the following minor comments to this work:

In this study, Vitam D supplementation did not improve muscle function in mdx mice while improved force production and fatigue resistance in control animals. Putative reasons beyond the lack of vitamin D effects in mdx mouse are well described in the discussion section, however the lack of specific systemic (e.g., serum Vitamin D quantification) and molecular analyses in the dystrophic EDL or soleus muscle (e.g., fibre type composition, vitamin D receptor content, calcium content) limit the possibility of drawing causal conclusions. Even if these molecular investigations were beyond the scope of this paper, the addition at the end of the discussion of possible future molecular analyses to be carried out would make this paper more complete.

INTRODUCTION

-The introduction is detailed and well-articulated, yet not overly cumbersome. Please include study hypotheses as part of the introduction.

-To highlight the important role played by vitamin D in muscle function authors mention only vitamin D deficiency in elderly (e.g., page 2 lanes 50-53), my suggestion is also to include other conditions were vitamin D-deficiency might impact muscle function/quality. For example the following review described the beneficial role of Vitamin D in postmenopausal-associated sarcopenia: Agostini D, Zeppa Donati S, Lucertini F, Annibalini G, Gervasi M, Ferri Marini C, Piccoli G, Stocchi V, Barbieri E, Sestili P. Muscle and Bone Health in Postmenopausal Women: Role of Protein and Vitamin D Supplementation Combined with Exercise Training. Nutrients. 2018 Aug 16;10(8). pii: E1103. doi: 10.3390/nu10081103).

Page 1, lane 42: please check reference format: “….dystrophin [12] [13]”.

Page 2, lane 50: A reference should be added at the end of the phrase: “….type II muscle fibres”.

RESULTS

Page 6, lane 190: please remove the point after the word “…..pseudohypertrophy.”

Author Response

We thank the reviewer for their constructive comments. In relation to their advice, the following changes have been applied:

Hypotheses have been added to the end on the introduction – ‘It was hypothesised VitD supplementation would have several effects on mdx dystrophic muscle including increased absolute force production, decreased fatigue following repeated muscle stimulation, increased recovery capacity post fatigue and enhanced muscle regeneration.’

Reference to vitamin D affecting other conditions such as post-menopausal associated sarcopenia and prostate cancer have been added

Page 1, lane 42: reference format fixed

Page 2, lane 50: reference has been added (Lynch, G.S. et al. The Journal of Physiology 535:, 591-600, 2001)

Page 6, lane 190: point removed after ‘pseudohyertrophy’

Reviewer 2 Report

The description of the methods seemed adequate for the process but a bit more clarity with relationship of the mdx mouse at the level of treatment introduction could be clarified with rationale for that.  

With regard to the Chow - the only change in the feedings was the level of Vit D - right?  This might be improved with some attention paid to the English grammar - L 123 i believe data are plural not singular; it seems more clear if the that is eliminated from the phrase 'that there' L192; couple of other places , the sentences could be more clearly state.

More clarity would be useful into how this aspect of assessment in the mdx mouse  with this condition differs from the aging process as well as the injured muscle repair process.

Author Response

We thank the reviewer for their constructive comments. In relation to their advice, the following changes have been applied:

Clarity as to the state of the mdx mouse at 8 weeks of age when treatment was applied  has been added to the methods - ‘Animals were chosen at this age to avoid the peak damage/regeneration period that occurs from 3-6 weeks of age, thus aligning with a period of slower progression of the disease occurring during a period of animal growth and maturation.’

You are correct. To avoid any confusion, a specific statement  highlighting that vitamin was the only difference in the diets was added to the methods – ‘Other than the differing vitamin D levels, the three diets were identical.’

Data now reads ‘The data is …’ and the word ‘that’ was deleted

We deliberately decided not to include a comparison of aging and injury/repair and the mdx mouse to focus on vitamin D as a potential therapy. This was done as it would depend on the age of the mouse, muscles chosen for analysis, as well as the extent of the aging process, and thus would have  expanded the size of the paper considerably, as well as deflecting the focus. Muscle repair as a process is not majorly different in dystrophic muscle, just the extent of it means repair eventually fails and the fat infiltration that occurs may be a link between sarcopenia (as opposed to healthy aging). However we have added some reference to other conditions such as sarcopenia and prostate cancer that vitamin D have been shown to be effective in.

Reviewer 3 Report

In my opinion this is a high quality paper.

However, I have some minor recommendations that I think should be done before publishing it in this journal.

- In the Introduction, I think it would be interesting to deepen in the possible applications of this paper to sport and physical activity.

- At the end of the Introduction, after the objective of the paper, I suggest to add the hypothesis of the study.

- In Statistical Analysis, please add effect size measures.

- In the Discussion, I would explain with more details the reasons why the authors think that VitD supplementation did not appear to have a significant impact on either contractility or regeneration in mdx mouse muscles, especially if that was the hypothesis of the study. 

- At the end of the discussion, possible future lines of research should be explained with detail considering the results of this study.

- In Conclusions, I consider really interesting the last sentence: These findings could suggest that VitD supplementation may be beneficial from an athletic point of view, thus providing further avenue for future research. 

¿beneficial in what sense? ¿what kind of future research? 

- In references, the dois should appear always in the same format.

Author Response

Reviewer 3

We thank the reviewer for their constructive comments. In relation to their advice, the following changes have been applied:

As the manuscript is specifically investigating the potential of vitamin D as a therapy for muscle wasting, we do not consider expanding the introduction to sport/physical activity is appropriate. However, as we did observe beneficial effects in the normal animals muscle function, we have expanded the paragraphs at the end of the discussion to highlight some of the possible effects of vitamin D related to sport/physical activity.

Hypotheses have been added to the end of the introduction

We have investigated the effect sizes and 95% confidence intervals, and these are consistent with the significance values that already exist in the paper (ie the significant differences are large effect sizes, while any that are not quite significant also only demonstrate small effect sizes). Thus we do not consider descriptions/discussion of these adds significantly to the manuscript. However, if desired, the figures showing the confidence intervals can be included as supplementary information.

In the discussion, we have already attempted to explain why the vitamin D was ineffective, including lack of changes in calcium handling, altered fibre type and the fact the dystrophic animals were unlikely to be vitamin D deficient when the study began. We have added the following ‘Alternately, if VitD were decreasing the number of degenerating muscle fibres, whilst also increasing the rate of regeneration, as suggested by [refs] then the overall outcome in centralised nuclei could be the same and may conceal any effects of VitD.’ to help explain the lack of effect in regeneration. While we could further consider aspects such as vitamin D receptor, vitamin D binding protein or the activity of vitamin D metabolic enzymes, this would have been pure speculation and thus would not add significantly to the manuscript.

At the end of the discussion we have added further discussion regarding the links of vitamin D and exercise performance  and included ‘Thus, future studies that combine high dose VitD supplementation and exercise in healthy models should be considered.’ as a final a statement for possible future research.

The conclusion simply alludes to the fact that this may be a possibility, and we have added ‘to increase muscle performance when applied in a vitamin D replete state’ to clarify the conditions under which this may be examined.

Reference doi’s have been updated to same format